# Design of Smart Nanodiamonds: Introducing pH Sensitivity to Improve Nucleic Acid Carrier Efficiency of Diamoplexes

**DOI:** 10.3390/pharmaceutics14091794

**Published:** 2022-08-26

**Authors:** Saniya Alwani, Raj Rai, Isabella Zittlau, Jonathan Rekve, Deborah Michel, Ildiko Badea

**Affiliations:** College of Pharmacy and Nutrition, University of Saskatchewan, Saskatoon, SK S7N 5A9, Canada

**Keywords:** nanodiamonds, amino acids, plasmid DNA, siRNA, endo-lysosomal escape, flow cytometry, peptide chemistry, proton sponge effect

## Abstract

The mechanism of cellular uptake and intracellular fate of nanodiamond/nucleic acid complexes (diamoplexes) are major determinants of its performance as a gene carrier. Our group designed lysine-nanodiamonds (K-NDs) as vectors for nucleic acid delivery. In this work, we modified the surface of K-NDs with histidine to overcome endo-lysosomal entrapment diamoplexes, the major rate limiting step in gene transfer. Histidine is conjugated onto the NDs in two configurations: lysyl-histidine-NDs (HK-NDs) where histidine is loaded on 100% of the lysine moieties and lysine/lysyl-histidine-NDs (H_50_K_50_-NDs) where histidine is loaded on 50% of the lysine moieties. Both HK-NDs and H_50_K_50_-NDs maintained the optimum size distribution (i.e., <200 nm) and a cationic surface (zeta potential > 20 mV), similar to K-NDs. HK-NDs binds plasmid deoxyribonucleic acid (pDNA) and small interfering ribonucleic acid (siRNA) forming diamoplexes at mass ratios of 10:1 and 60:1, respectively. H_50_K_50_-NDs significantly improved nucleic acid binding, forming diamoplexes at a 2:1 mass ratio with pDNA and a 30:1 mass ratio with siRNA, which are at values similar to the K-NDs. The amount of histidine on the surface also impacted the interactions with mammalian cells. The HK-NDs reduced the cell viability by 30% at therapeutic concentrations, while H_50_K_50_-NDs maintained more than 90% cell viability, even at the highest concentrations. H_50_K_50_-NDs also showed highest cellular uptake within 24 h, followed by K-NDs and HK-NDs. Most functionalized NDs show cellular exit after 5 days, leaving less than 10% of cells with internalized diamonds. The addition of histidine to the ND resulted in higher transfection of anti-green fluorescent protein siRNA (anti-GFP siRNA) with the fraction of GFP knockdown being 0.8 vs. 0.6 for K-NDs at a mass ratio of 50:1. H_50_K_50_-NDs further improved transfection by achieving a similar fraction of GFP knockdown (0.8) at a lower mass ratio of 30:1. Overall, this study provides evidence that the addition of histidine, a pH-modulating entity in the functionalization design at an optimized ratio, renders high efficiency to the diamoplexes. Further studies will elucidate the uptake mechanism and intracellular fate to build the relationship between physicochemical characteristics and biological efficacy and create a platform for solid-core nanoparticle-based gene delivery.

## 1. Introduction

Gene therapy is a promising direction targeting incurable cancer [1] and debilitating medical conditions [2,3,4]. It also facilitates the development of novel vaccines against life-threating infections [5,6] recently in the spotlight for the success of genetic vaccines against SARS-CoV-2 virus [7,8,9]. The utilization of therapeutic nucleic acids (DNA and RNA) requires sophisticated carriers capable of packing, protecting, transporting, and delivering them to target cells in intact form. These carriers should not only have optimum affinity for the genetic material but should also be non-toxic. Despite intense research in this field over many years, only a few nanoparticle (NP)-based genetic treatments have reached clinical translation [10,11,12]. Evidently, there is an unmet need to develop new delivery platforms capable of overcoming the challenges we face in this arena.

Among many non-viral gene-delivery vectors, carbon NPs have a unique standing [13,14,15,16], with nanodiamonds (NDs) being the most biocompatible members of this family [17,18]. Moreover, NDs resist biodegradation, present a tunable surface chemistry to conjugate multiple chemical moieties, and have a high surface-to-volume ratio, allowing loading of genetic material in higher concentrations [19]. However, small NDs (5 to 10 nm) tend to form aggregates of tens to hundreds of nanometers to minimize their surface free energy. The propensity of NDs to aggregate in tight structures is a formulation challenge for biological applications [20]. It compromises dispersion stability, reproducibility, loading capacity, cellular uptake and in vivo biological safety and is thus a major focus of research groups exploring NDs as nanocarriers [21,22].

Our group covalently functionalized NDs for the first time with a basic amino acid, lysine, to impart a net-positive charge on the surface capable of forming ionic complexes with negatively charged nucleic acids [23]. Previously, we showed that our functionalization achieved surface loading ranging from 1.67 to 1.97 mmoles/g [15,23], significantly higher than previously reported in the literature to covalently attach alkyl groups on NDs [24]. The lysine-NDs (K-NDs) showed minimum aggregation and maintained optimum dispersion stability for the testing period of 25 days. K-NDs formed diamoplexes with plasmid deoxyribonucleic acid (pDNA) and siRNA at 1:1 and 20:1 mass ratios and showed a dose-dependent cellular uptake in two mammalian cancer cell lines [15].

Despite good binding with nucleic acids, biocompatibility, cellular uptake and translocating siRNA intracellularly [15], K-NDs showed a suboptimal transfection efficiency [25]. We geared our evaluations to identify intracellular barriers for efficient gene transfer [26]. Diamoplexes internalized mammalian cells using two mechanisms, clathrin-mediated endocytosis and macropinocytosis. Intra-cellular mapping revealed that upon cellular entry, the diamoplexes remained entrapped in endo-lysosomal compartments, risking pre-mature degradation of the genetic cargo [26]. Therefore, modifications were considered to equip K-NDs for escaping endosomal trafficking, so that diamoplexes can exist freely in the cytoplasm for gene transfer.

Some NPs are innately capable of triggering escape mechanisms without secondary interventions. For example, calcium phosphate (Ca_2_PO_4_) composites rapidly dissolve upon entering the acidic milieu of the endosomes, increasing the osmotic pressure within the organelle. This pressure ultimately ruptures the membrane, releasing NP/nucleic acid complexes in the cytosol [27]. Carbon nanotubes may undergo an endocytosis-free entry into the cells—owing to their morphology—by simply penetrating through the cell membrane as nano-needles, ensuring high transfection efficiency but also high toxicity secondary to off-target cellular interactions [28].

Not all NPs are innately capable of escaping endo-lysosomal trafficking; thus, many secondary mechanisms are needed to be built into their structure to overcome this barrier. Some chemicals, such as cationic peptides, work by creating pores in the endosomal membrane, making these organelles leaky [29]. Fusogenic lipids form neutral ion pairs with the anionic endosomal membrane, disrupting the packing and releasing encapsulated biomolecules [29]. UV/visible-light-mediated photonic disruption is also evaluated by attaching a photosensitive mediator such as meso-tetraphenylporphine to NPs [29]. However, the most common approach to facilitate endosomal escape of nanocarriers is to disrupt endosomal membrane via the proton sponge effect [29]. It is mediated via moieties with a high buffering capacity and the ability to protonate at a lower pH. Polymers such as polyamidoamine (PAMAM) and polyethyleneimine (PEI) were considered gold standards to induce the proton sponge effect due to the presence of titratable amines in their chains [30,31,32]. Over the years, more complex molecules are designed by combinatorial chemistry, where polymers are combined with other pH-buffering agents, mainly histidine. One hybrid model combines a histidine-rich peptide LAH4-L1 with PAMAM and sleeping beauty transposon plasmid system, showing more than 90% gene transfection efficiency in cancer cells [33]. Another example is a multifunctional polymeric carrier 1,4,7-triazanonylimino-bis[N-(oleicyl-cysteinyl-histinyl)-1-aminoethyl) propionamide] containing peptides rich in histidine to allow conjugation with siRNA and efficient endosomal escape [34]. Endo-Porter, a synthetic histidine-rich, pH-dependent membrane-active peptide, is also designed, where leucine residues help in binding with the endosomal membrane and the histidine residues disrupt them simultaneously [35]. All these examples build evidence that histidine is a mediator of endosomal escape. However, in most cases, it is incorporated in complex chemical configurations such as peptides [34,36]. Utilization of histidine in a simpler configuration on the surface of NDs might bring additional benefits.

To improve the endosomal escape of K-NDs, we introduced histidine on the ND surface by covalently conjugating it to the lysine residues. Upon encounter with the acidic medium of the endosomes, histidine undergoes protonation, leading to a cascade of counter ion and water influx in the organelle to regain the osmotic balance. This influx leads to early endosomal swelling, membrane rupture and release of content in the cytosol before reaching the lysosomal stage, thus preventing degradation of the genetic material in the lysosomes [37,38].

However, this modification should not affect the physicochemical characteristics, nucleic acid binding and cellular uptake of the K-NDs [15,23]. Considering this important pre-requisite, we added histidine in two configurations: (1) lysyl-histidine-NDs (HK-NDs) where histidine is covering 100% of the surface as a di-peptide with lysine covalently attached to surface carboxylic groups of NDs, and lysine/lysyl-histidine NDs (H_50_K_50_-NDs) where histidine covers 50% of the surface as di-peptide while the remaining 50% is covered with lysine alone both attached covalently to carboxylic groups on the NDs. The work presented below provides a systematic comparison of physicochemical properties, gene binding and cellular interactions of these new constructs, evidencing that the addition of histidine as a pH-sensitive moiety improves the overall performance of the carrier.

## 2. Materials and Methods

### 2.1. Materials

Pharmaceutical-grade (ND98) carboxylic acid–functionalized NDs with an average particle size of 5 nm were purchased from Dynalene Inc. (Whitehall, PA, USA). YTZ^®^ (Yttrium Stabilized Zirconia) grinding media (0.05 mm) were purchased from Tosoh Corporation (Grove City, OH). Details of purchase for chemicals including Nα-boc-L-lysine-OH (purity 97.5%), Nα,Nim-bis-boc-L-histidine-hydroxysuccinimide ester (purity ≥ 98%), fmoc-1,3-diaminopropane hydrochloride (purity ≥ 99%), N,N’-Diisopropylethylamine (purity ≥ 99%; DIPEA), piperidine (purity ≥ 99.5%), and N,N,N′,N′-tetramethyl-O-(7-azabenzotriazol-1-yl)uroniumhexafluorophosphate (purity ≥ 99.5%; HATU) used in the synthesis of HK-NDs and H_50_K_50_-NDs were published earlier [15,23,39]. Molecular-biology-grade agarose (CAS Number: 9012-36-6) was purchased from Millipore Sigma (Oakville, ON, Canada). HyClone^®^ HyPure Molecular Biology Grade Water, HyClone™ 1x Phosphate Buffered Saline (1xPBS) and HyClone™ Dulbecco’s Modified Eagle’s Medium (DMEM)/High; glucose with L-glutamine and sodium pyruvate was obtained from ThermoFisher Scientific (Waltham, MA, USA). Fetal bovine serum (FBS) was acquired from Gibco (Waltham, MA, USA). Scrambled control siRNA and Anti GFP siRNA, Silencer^®^ GFP (eGFP) siRNA (50 µM, 5 nmoles; catalogue number AM4626) were purchased from Ambion, (Waltham, MA, USA). pDNA was prepared in house. Lipofectamine RNAimax (catalogue number 13778-150) was also purchased from Invitrogen (Waltham, MA, USA). Human cervical cancer cells (HeLa cells) and HeLa/GFP cells (HeLa cells transfected with green fluorescent proteins) were obtained from American Type Culture Collection (Manassas, VA, USA). Trypsin (0.25%; catalogue number: 25200-056) and antibiotic/anti-mycotic solution 100× (catalogue number: 15240-062) were obtained from Sigma Aldrich (Oakville, ON, Canada). MTT (3-(4,5-Dimethylthiazol-2-yl)-2,5-Diphenyltetrazolium Bromide) (catalogue number: M6494) was purchased from Invitrogen (Waltham, MA, USA) and was prepared as 5 mg/mL solution in house using 1XPBS as the diluent.

### 2.2. Synthesis of HK-NDs and H_50_K_50_-NDs

Pristine NDs were subjected to reoxidation using a combination of strong acids (nitric acid and sulfuric acid at 1:3 volume ratio) to ensure that all functional groups on the ND surface were converted into carboxylic acids (COOH), yielding re-oxidized NDs (rNDs). The synthesis of K-NDs was reported earlier [23]. Quantification of surface functionalization was also previously reported [15,23]. Two new functionalization protocols are addressed in this paper (Figure 1):

#### 2.2.1. Synthesis of HK-NDs

HK-NDs where histidine was attached to lysine, and this di-peptide conjugate covalently bonded to rNDs via an amine-terminated three-carbon chain linker. The detailed protocol to synthesize HK-NDs is presented elsewhere [39]. Briefly, Nα, Nim-bis-boc-L-histidine-hydroxysuccinimide ester (8.66 mmol), HATU (mole ratio of histidine/HATU = 1:0.9; 7.79 mmol) and DIPEA (mole ratio of histidine/DIPEA = 1:2; 17.97 mmol) were dissolved in 15 mL of DMF and stirred in a nitrogen gas-stabilized Schlenk flask for 15 min. N_α_-boc-L-lysine-OH (mole ratio of histidine/lysine = 1:1; 8.66 mmoles) dissolved in 10 mL of DMF was added to the reaction mixture and kept under constant stirring for 18 h to form N_ε_-(N_α_,N_im_-bis-boc-histidine)-N_α_-(boc)-lysine (compound **1**). Thereafter, 3 mmoles of compound **1** dissolved in 15 mL of DMF was added to the reaction mixture containing the linker fmoc-1,3-diaminopropane (mole ratio of amino acid conjugate/linker = 1:1.5; 9.92 mmol), HATU and DIPEA dissolved in DMF to form N_ɛ_-(N_α_-N_im_-bis-boc-histidine) N_α_-boc-lysyl-fmoc diaminopropane (compound **2**). Deprotection of the fmoc protecting group from compound **2** was carried out using 50% (*v*/*v*) piperidine-dry DCM mixture forming N_ɛ_-(N_α_-N_im_-bis-boc-histidine) N_α_-boc-lysyl-diaminopropane (Compound **3**). rNDs were treated with excess thionyl chloride in the presence of YTZ grinding media, forming acyl chloride functionalized NDs. Compound **3** was attached to acyl chloride functionalized NDs, following which Boc deprotection was carried out using 4 M HCl in dioxane to yield HK-NDs. NDs were lyophilized for storage. This step was replicated from a previously published protocol [23].

#### 2.2.2. Synthesis of H_50_K_50_-NDs

H_50_K_50_-NDs where the carboxylic acid surface of rNDs was conjugated with a 50–50 molar ratio of lysine and lysyl-histidine peptide via an amine-terminated three-carbon chain linker used for the K-NDs. The surface loading of rNDs (1.0 mmole/g) reported earlier [23] was used to calculate the amount of lysine and lysyl-histidine needed to achieve a 50/50 configuration. Both moieties for attachment were protected as previously described [23,39]. At first, 0.5 mmoles of lysine was added followed by 0.5 mmoles of lysyl-histidine 15 min later. This was performed to ensure that lysine attached to the surface efficiently before introducing a bulkier functional group, thus avoiding stearic hinderance.

### 2.3. Size and Zeta Potential Measurements

An aqueous dispersion of HK-NDs was prepared at a 2 mg/mL concentration. Dispersions of H_50_K_50_-NDs were first made at concentration of 2 mg/mL, but after preliminary analyses dilutions were performed to 1.5 mg/mL and 1 mg/mL to optimize the formulation for biological experiments. All ND dispersions were subjected to ultrasonication for 12 h at 25 kHz without heating in the presence of YTZ grinding medium at a ratio of 1:1. The dispersions were then centrifuged at 5200× *g* for 5 min to sediment the YTZ grinding medium. Particle size distributions were obtained using Malvern Zetasizer Nano ZS instrument (Malvern Instruments Ltd., Malvern, Worcestershire, UK). Solvent properties were as follows: refractive index = 1.330, dielectric constant = 78.5 and viscosity at 25 °C = 0.8872 cP. Using the CONTIN algorithm, the decay rates as a function of the translational diffusion coefficients of the particles ‘D’ were analyzed. The hydrodynamic radius RH of the particles was then calculated using the Stokes–Einstein equation (RH = kT/6πηD), where k = Boltzman constant, T is the temperature (25 °C), and η = viscosity of water at 25 °C. The calculated particle size was an estimate based on the hydrodynamic radius of spherical particles with a translational diffusion coefficient equal to the particles in the dispersion. All the size-distribution readings were derived from three measurements. Each measurement had a minimum of ten individual runs. The resulting data were reported as volume distribution. Zeta potential measurements were performed using Doppler electrophoresis and phase analysis. All the reported zeta potentials were the average of three measurements. Each measurement had a hundred individual runs.

### 2.4. Transmission Electron Microscopy (TEM)

TEM was conducted to evaluate the morphology of ND clusters resulting from various surface functionalizations. Aqueous dispersions of rNDs and fNDs were prepared as described above and were further diluted up to two-fold. A 10 μL droplet of dispersion was applied on carbon film-coated copper grids then incubated for 5 min, and excess liquid was wicked off with absorbent tissue from the edge of the grid. All samples were dried at ambient temperature for 24 h and imaged using a Hitachi HT7700 transmission electron microscope at 100 keV.

### 2.5. Gel Electrophoresis

Aqueous dispersions of HK-NDs and H_50_K_50_-NDs were prepared at 2 mg/mL and 1 mg/mL concentrations, respectively. A 1% agarose gel was prepared in tris-acetate ethylenediaminetetraacetic acid (TAE) buffer with 0.01% ethidium bromide. Diamoplexes were prepared at mass ratios ranging from 2:1 to 50:1 (NDs: pDNA) by incubating pDNA with ND dispersions for 30 min at room temperature. Finally, the samples were loaded in the gel with 3% glycerol solution in water and subjected to electrophoresis at 100 V for 80 min using Bio-Rad PowerPac HC electrophoresis apparatus (Bio-Rad Laboratories, Tnc. Mississauga, ON, Canada). The gel was then imaged using an AlphaImager imaging system (Alpha Innotech Corporation, San Leandro, CA) to detect fluorescence. The amount of pDNA in each sample was 500 ng and free pDNA was loaded as the negative control.

Binding efficiency with siRNA was also analyzed using the same concentrations for both HK-NDs and H_50_K_50_-NDs. A 2% agarose gel was prepared in TAE buffer with 0.01% ethidium bromide. Diamoplexes were prepared at mass ratios ranging from 20:1 to 160:1 for HK-NDs and 10:1 to 140:1 for H_50_K_50_-NDs (ND: siRNA) by incubating the siRNA with ND dispersions for 30 min at room temperature. Electrophoresis was run for 50 min at 75 V and the samples were analyzed the same way as the pDNA complexes. The amount of siRNA in each sample was 400 ng and free siRNA was used as the negative control.

### 2.6. Formulation Development for In Vitro Assays

Figure 2 depicts a schematic illustration of ND formulation development in vitro cellular assay. Briefly, primary dispersions were prepared in molecular-biological-grade deionized water at the following concentrations: 2 mg/mL (rNDs, K-NDs and HK-NDs) and 1 mg/mL (H_50_K_50_-NDs). YTZ grinding medium was added in all dispersions at a ratio of 1:1. All primary dispersions were subjected to ultrasonication for 4 h at a frequency of 25 kHz without heating. The ultrasonicated dispersions were then centrifuged at 5200× *g* for 5 min to sediment the YTZ grinding medium. All fND dispersions were diluted with serum-supplemented cell culture media (s-DMEM) to final concentrations required for the bioassays. Diamoplexes were prepared by incubating the dispersions of NDs with siRNA (target siRNA = anti-GFP siRNA also called Silencer™ GFP (eGFP) siRNA and control siRNA = scrambled siRNA, also called Negative Control #1 siRNA) at room temperature for 30 min. Attained diamoplexes were diluted with s-DMEM to final working concentrations.

### 2.7. MTT-Cell Viability Assay

The biocompatibility of HK-NDs and H_50_K_50_-NDs was assessed using MTT (3-(4,5-Dimethylthiazol-2-Yl)-2,5-Diphenyltetrazolium Bromide) assay in cervical carcinoma cells (HeLa cells) both alone and as diamoplexes. rNDs and K-NDs were assessed from concentrations of 10 µg/mL to 250 µg/mL, while HK-NDs and H_50_K_50_-NDs were assessed up to 1000 µg/mL and 500 µg/mL, respectively, based on their differential affinities to nucleic acids. Nevertheless, in all cases, the maximum tested concentration between 3- and 7-fold higher than the concentration needed for functional assays. The biocompatibility of all functionalized NDs (fNDs) was also assessed as diamoplexes with siRNA made at a 20:1, 50:1 and 30:1 weight ratio for K-NDs, HK-NDs and H_50_K_50_-NDs, respectively. Each sample contained 5 pmoles of siRNA/well containing 5000 cells.

For all samples, the cells were plated in a 96-well plate at a density of 5000 cells per well and incubated at 37 °C and 5% CO_2_ to allow attachment to the plate surface for 24 h. The cells were then treated with NDs at the respective concentration, keeping the final volume of treatment at 100 µL using s-DMEM as the diluent. After 24 h, the treatments were terminated, and the cells were washed with 1XPBS three times each. A measure of 10 µL of MTT stock solution (5 mg/mL of MTT in PBS) diluted in 90 µL of s-DMEM was added in each well, and the cells were incubated at 37 °C/5% CO_2_ for one hour. Thereafter, the supernatant was removed, and formazan crystals resulting from the mitochondrial reduction of MTT were dissolved using 100 µL of dimethyl sulfoxide (DMSO). The plates were incubated for another 10 min to remove air bubbles. The absorbance was read at 550 nm using a microplate reader (BioTek^®^ Microplate Synergy HT, VT, USA). Untreated cells were used as the control. The percent cell viability was calculated using the following formula:$$\%\;\mathrm{cell}\;\mathrm{viability}=\frac{{\mathrm{Absorbance}}_{\mathrm{treated}}}{{\mathrm{Absorbance}}_{\mathrm{control}}}\times 100$$

### 2.8. Flow Cytometry Analysis to Compare Cellular Entry and Exit Profile

In order to compare the magnitude of cellular uptake and cellular exit of histidine modified ND constructs to the prototype, we utilized flow cytometry as published earlier [15]. The following samples were included in the experiment: (a) untreated cells (b) K-NDs treated cells, (c) HK-NDs treated cells, and (d) H_50_K_50_-NDs treated cells analyzed at 24, 48 and 120 h post-treatment.

HeLa cells were plated on a 24-well plate at a density of 80,000 cells/mL and incubated at 37 °C and 5% CO_2_ for 24 h to allow attachment. Thereafter, the cells were subjected to respective treatments with various fNDs at a concentration of 150 µg/mL. All treatments were terminated exactly after 24 h followed by washing with PBS three times for 5 min each. Sample analysis was performed after 24, 48 and 120 h of treatment removal. Samples for the 120 h time point were replated after 72 h in fresh 24-well plate and incubated again 37 °C and 5% CO_2_ with s-DMEM for another 48 h. This was carried out to restrict the growth cycle and inhibit cell multiplication in order to maintain equivalent cell counts across all samples. The average doubling time of Hela cells is reported to be 33.7 ± 6.7 h [40]; therefore, replating of 120 h samples on day 3 was ideal. For analysis, samples were harvested using 0.25% trypsin in ethylenediaminetetraacetic acid (EDTA) solution and centrifuged at 95× *g* and 4 °C for 5 min to obtain cell pellets. Individual cell pellets were resuspended in 500 µL of PBS and transferred into 5 mL flow cytometer tubes.

Analyses were performed on CyFlow Space (Partec GmBH, Munster, Germany) equipped with a 400 mW argon laser, and the data were acquired by FloMax software (version 2.4) provided by Partec GmbH. Scattering of light by the cells at narrow angles to the axis of the laser beam, i.e., forward scattering (FSC), and at an approximately 90° angle to the axis of the laser beam, i.e., side scattering (SSC), was measured as a parameter of analysis. FSC corresponds to changes in the overall diameter of the analyzed cell while SSC corresponds to the changes in granularity (internal complexity) secondary to light-refracting intracellular moieties [41]. For each sample, 10,000 total events were collected from the healthy cell population. The cell population was divided in four quadrants using the untreated cells as the reference to quantify changes in FSC and SSC intensities. Cells in QA3 were considered to have no/the least NDs while cells in QA1 were considered to have NDs as refracting entities. To compare the magnitude of cellular uptake for various fNDs, the shift from QA3 to QA1 was assessed.

### 2.9. Flow Cytometry Analysis of siRNA-Mediated Green Fluorescent Protein (GFP) Knockdown

Flow cytometry was used for assessment of the gene knock-down efficiency of fNDs. Diamoplexes of K-NDs and HK-NDs were prepared as described above at mass ratios of 50:1 (ND: siRNA), while H_50_K_50_-NDs were evaluated at mass ratios of 30:1, 50:1 and 70:1 (ND:siRNA). HeLa/GFP cells (GFP expressing cells) were incubated with diamoplexes containing anti-GFP siRNA for 48 h at 37 °C and 5% CO_2_. Cells treated with diamoplexes of scrambled siRNA were used as the negative control while cells treated with anti-GFP siRNA/lipofectamine RNAimax-based lipoplexes were used as the positive control. The final concentration of anti-GFP siRNA in each well was 75 pmoles. Upon treatment termination, all cells were harvested and dispersed in 500 µL of PBS for analysis via BD FACS Calibur™ (BD Biosciences, San Jose, CA, USA). BD CellQuest™ Pro software was used for data analysis. As the emission wavelength of GFP is 509 nm, the fluorescence of cells was measured using an FL1 (fluorescence intensity channel 1) filter, i.e., the channel of flow cytometry capable of capturing an emission wavelength in 530 ± 30 nm. Approximately 13,000 events were recorded for each sample in the gated region (gate was applied to exclusively capture the healthy cell population). In order to analyze the negative shift in GFP fluorescence, the histograms representing GFP intensity were divided into four regions (no-GFP, low-GFP, mid-GFP and high-GFP). Comparison of GFP knockdown was performed by quantifying the shift in the cell population from high- and mid-GFP regions to low- and no-GFP regions. Percent GFP knockdown was calculated as follows:$$\mathrm{Percent}\;\mathrm{GFP}\;\mathrm{knockdown}\;\left(\%\right)=\left[1-\frac{\mathrm{Percent}\;\mathrm{GFP}\;\mathrm{fluorescence}\;\mathrm{in}\;\mathrm{low}\;\mathrm{to}\;\mathrm{no}\;\mathrm{GFP}\;\mathrm{region}\;\mathrm{of}\;\mathrm{untreated}\;\mathrm{cells}}{\;\mathrm{Percent}\;\mathrm{GFP}\;\mathrm{fluorescence}\;\mathrm{in}\;\mathrm{low}\;\mathrm{to}\;\mathrm{no}\;\mathrm{GFP}\;\mathrm{region}\;\mathrm{of}\;\mathrm{treated}\;\mathrm{cells}}\right]\times 100$$

As fNDs were analysed in different runs and instruments, the knock-down was standardized to the positive control, lipofectamine. The fraction of GFP knockdown was calculated for each type of fNDs with respect to the lipofectamine RNAimax-based lipoplexes using the following formula:$$\mathrm{Fraction}\;\mathrm{of}\;\mathrm{GFP}\;\mathrm{knockdown}\;\mathrm{with}\;\mathrm{respect}\;\mathrm{to}\;\mathrm{lipofectamine}\;\mathrm{control}\;\;\;\;\;\;\;\;\;\;\;\;\;\;\;\;\;\;\;\;\;\;\;\;\;\;\;\;\;\;\;\;\;\;\;\;\;\;\;\;\;\;\;\;\;\;\;\;\;\;=\frac{\mathrm{Percent}\;\mathrm{GFP}\;\mathrm{knockdown}\;\mathrm{of}\;\mathrm{diamoplex}\;\mathrm{treated}\;\mathrm{cells}}{\mathrm{Percent}\;\mathrm{GFP}\;\mathrm{knockdown}\;\mathrm{of}\;\mathrm{lipoplex}\;\mathrm{treated}\;\mathrm{cells}}$$

### 2.10. Statistical Analysis

All results were expressed as the mean of *n* ≥ 3 ± S.D. Levene’s test (*p* > 0.05) was applied to test the homogeneity of variance. ANOVA and Tukey’s post hoc multiple comparisons were used for comparing the pairs. The significant differences were considered at a *p* ≤ 0.05 level of significance (α).

## 3. Results

### 3.1. Physicochemical Properties of HK-NDs and H_50_K_50_-NDs

Both HK-NDs and H_50_K_50_-NDs were initially prepared for analysis as 2mg/mL dispersions, similarly to the K-NDs reported earlier [15]. In contrast to K-NDs, which showed 89% of particles within the range of 30–90 nm, the HK-NDs showed 56% of the population in this size range. Approximately 31% of the particles were in 100 to 200 nm while 9% were in the 200–300 nm range (Table 1). The final dispersion showed no visible aggregation (polydispersity index (PDI) = 0.2). Therefore, stock dispersions of HK-NDs for subsequent biological experiments were prepared at a concentration of 2 mg/mL. The H_50_K_50_-NDs prepared at a 2 mg/mL concentration showed poor stability and a high degree of aggregation. Most particles, i.e., 97%, were >1000 nm, and were thus unsuitable for biological application. To improve the formulation stability, we diluted the dispersion of H_50_K_50_-NDs to 1.5 mg/mL. This step improved particle size distribution yielding 52% of particles < 1000 nm. However, the system was unable to show uniformity (PDI = 0.4) and still exhibits a high degree of aggregation (49% particles > 1000 nm). Further dilution of H_50_K_50_-ND dispersion to 1 mg/mL significantly improved the dispersion quality as most of the particles remained below 500 nm (80%) with 48% being below 200 nm, thus exhibiting reduced aggregation. Only 8% of the particles existed as aggregates of more than 1000 nm. The final dispersion of H_50_K_50_-NDs at 1 mg/mL presented good uniformity with a PDI of 0.2 (Table 1).

Electron micrographs show particle size range consistent with DLS measurements of 200–500 nm. Both HK-NDs and H_50_-K_50_-NDs exist in clusters of primary diamond crystals showing clear particle boundaries, thus suggesting loose aggregation and easy dispersibility (Figure 3).

Unlike particle size distributions that considerably changed following the addition of histidine on the system, zeta potentials of both new fNDs remained close to the prototype. Like K-NDs (zeta potential = 26–30 mV) [15]**,** the HK-NDs also possess a highly positive zeta potential of 30 mV (Table 1). For the H_50_K_50_-NDs, the zeta potential transitioned from 18 to 23 mV as the formulation concentration was optimized (Table 1). At a final concentration of 1 mg/mL, the H_50_K_50_-NDs showed similar zeta potential to the other NDs; thus, stock solutions are prepared with 1 mg/mL for subsequent biological experiments.

### 3.2. Nucleic Acid Binding Affinity of HK-NDs and H_50_K_50_-NDs

Assessment of binding affinity dictates the optimization of the functionalization approach for NDs. As depicted in Figure 4A, HK-NDs show binding with pDNA at a minimum mass ratio of 10:1 (ND:pDNA). This amount was double the amount of K-NDs needed for the same concentration of pDNA (5:1) [23]. At higher mass ratios of 25:1 and 50:1, complete disappearance of bands was evident, indicating strong complexation with the NDs (Figure 4A). H_50_K_50_-NDs showed pDNA binding at mass ratios of 2:1 and 5:1 (Figure 4B). At higher mass ratios ranging from 10:1 to 50:1, the pDNA band completely disappeared owing to the formation of strong ND/pDNA complexes (Figure 4B). It is of note that H_50_K_50_-NDs starts binding with DNA at lowest mass ratio of 2:1 compared to K-NDs (minimum mass ratio of 5:1) [23], thus causing a more than 2-fold decrease in the amount of NDs required to conjugate 500 ng of pDNA.

We also assessed the binding affinities of histidine-based fNDs with siRNA. Overall, HK-NDs show weaker binding with siRNA as compared to H_50_K_50_-NDs. At a 20:1 (ND: siRNA) mass ratio, a visible band was observed indicating an incomplete complexation (Figure 5A). Adequate complexation began at a mass ratio of 60:1 following a greater retention of siRNA in the diamoplexes. Complete disappearance was seen starting at a mass ratio of 100:1 owing to the formation of strong diamoplexes. However, in all wells, starting from 20:1, siRNA is also seen confined in the wells, indicating that complexation begins at the lowest mass ratio (Figure 5A). Considering that a balance between packaging and intracellular release of siRNA from these diamoplexes is required, the assay suggests that the optimum mass ratio required to deliver siRNA via HK-NDs should be 50:1, selected for the cellular assays. The H_50_K_50_-NDs started retaining the siRNA at a minimum mass ratio of 10:1 (Figure 5B). As the mass ratio increases, bands became lighter while more siRNA is seen confined in the wells, evidencing increasing interaction between H_50_K_50_-NDs and siRNA. A 30:1 ratio is a minimum mass ratio to form diamoplexes with H_50_K_50_-NDs. This value is comparable to the K-NDs which are able to bind siRNA at a minimum mass ratio of 20:1 [23], suggesting that reducing histidine on the ND surface to 50% in H_50_K_50_-NDs is favorable to achieve siRNA binding. Higher mass ratios were tested for histidine modified ND constructs as compared to the ones used for the K-NDs [23]. This was due to the fact that incomplete binding was observed at lower mass ratios.

### 3.3. Concentration Dependent Biocompatibility of fNDs

As indicated earlier, NDs are the most biocompatible among all carbon NPs [17,18]. Therefore, it is imperative to ensure that surface functionalization does not compromise the innately safe nature of diamond NPs. Hence, we established the concentration-dependent safety profile of NDs. Pristine NDs (as received from the supplier) and rNDs (NDs with a carboxylated surface) were compared to K-NDs, HK-NDs and H_50_K_50_-NDs (Figure 6). For each type of ND, the range of concentration was selected based on their binding affinities. However, the maximum concentration in every case was at least 10 times higher than the amounts needed for subsequent cellular assays.

The innate biocompatible nature of NDs was confirmed as pristine NDs maintained cell viability up to 90% even at the highest concentration of 250 µg/mL (Figure 6). After reoxidation, the biocompatibility of rNDs was slightly compromised as up to 87% and 78% cells were viable after exposure of 100 and 250 µg/mL concentration, respectively. K-NDs, having a cationic surface rich in primary amines, did not cause cytotoxicity, even at the highest concentration of 250 µg/mL, where more than 90% of the cells remained viable. These results provide evidence that rendering the surface of NDs cationic using lysine maintained the inert nature of the carrier. The addition of histidine on the surface significantly changed the profile, as the overall the HK-NDs exhibited higher toxicity compared to the K-NDs. Cellular viability after HK-ND treatment showed a dose-dependent decline from 75% at lowest the concentration of 10 µg/mL to 58% at the highest concentration of 1000 µg/mL (Figure 6). However, an anomaly is seen at 100 µg/mL where cell viability is reduced to 45%, i.e., higher than when cells are exposed to a 10-times-higher ND concentration of 1000 µg/mL, possibly due to experimental parameters. Unlike K-NDs wherein the cell viability remained unaffected by serum proteins, the HK-NDs were found to be more biocompatible in the absence of serum proteins (Appendix A).

In contrast to HK-NDs, the H_50_K_50_-NDs showed a biocompatibility profile comparable to K-NDs, maintaining more than 90% cell viability at the highest exposure of 500 µg/mL (Figure 6). It is to note that cells suffered less than 10% loss of viability at a concentration of 150 µg/mL employed for subsequent cellular assay. Thus, reducing 50% of histidine moieties from the surface restored the biocompatibility of the fNDs.

The biocompatibility of fNDs was also assessed as diamoplexes. Mass ratios to form diamoplexes were selected on the differential nucleic acid-binding ability of each type of fND: (a) diamoplexes of siRNA and K-NDs were prepared at a 20:1 mass ratio, (b) diamoplexes of siRNA and HK-NDs were prepared at a 50:1 mass ratio, and (c) diamoplexes of siRNA and H_50_K_50_-NDs were prepared at a 30:1 mass ratio (ND:siRNA).

The safety profile of diamoplexes prepared with the K-NDs and histidine-based constructs follow a similar pattern as the NDs alone. Diamoplexes made with K-NDs are the most biocompatible with a cell viability up to 97%. Diamoplexes of H_50_K_50_-NDs present a similar profile with 95% viability. In contrast to the above two, diamoplexes of HK-NDs reduced the cellular viability to 72% post-treatment (Figure 7). These diamoplexes were made at a 50:1 mass ratio; hence, the cells were exposed to ~37 µg/mL concentration of NDs. This is consistent with the data shown in Figure 6 that HK-NDs have the tendency to reduce cell viability to less than 75% at concentrations above 20 µg/mL.

### 3.4. Effect of Functionalization on Cellular Uptake and Exit of fNDs

Flow cytometric evaluations were performed (1) to compare the magnitude of cellular uptake of histidine-modified fNDs and (2) to ascertain elimination of NDs from the cells. Cells in QA3 were considered to have no NDs, thus showing minimal side scattering, while cells in QA1 have endocytosed NDs, shifting the cellular population upward due to higher side scattering. Side scattering in turn reflects increased internal complexity owing to internalized light-refracting diamond moieties and the presence of endosomes encapsulating these NDs. At each time point, the shift was normalized using untreated cells as they show similar patterns at every time point with approximately 95% of the cells in the QA3 region (Table 2 and Appendix A1,B1,C1). The H_50_K_50_-NDs show the highest SSC intensity with 48% of the cell population shifted to the QA1 quadrant, thus exhibiting the highest cellular uptake after 24 h post-treatment (Table 2 and Appendix A4), followed by K-NDs shifting 25% of the population to QA1 (Appendix A2), and HK-NDs showed the minimum SSC intensity with a 20% population shift to QA1 (Table 2 and Appendix A3). Comparing the SSC intensity and shift from the QA3 to QA1 region between K-NDs and H_50_K_50_-NDs, we can stipulate that the addition of histidine enhances the cellular uptake by at least two-fold. In contrast, comparing HK-ND to H_50_K_50_-NDs, it is evident that a balance is required between the amount of histidine and lysine on the surface of diamonds to mediate these cellular interactions. A 100% surface coverage with histidine in HK-NDs compromises the cellular uptake by more than twofold.

The second parameter analysed in this assay was the exit of NDs from mammalian cells as the function of their surface functionalization. Here, we treated the cells with various fNDs for 24 h, following which all treatments were terminated and cell samples were harvested in three batches: 24 h post-treatment, 48 h post-treatment, and 120 h post-treatment to compare the amounts of NDs remaining in the cytoplasm after each specific time point. A reduction in SSC intensities, shown as the downward shift of population from the QA1 to QA3 region, was used as a function to compare the number of internalized diamonds. The number of cells in the QA3 region remained unchanged (75 vs. 74% after 24 and 48 h, respectively) following K-ND treatments, showing that K-NDs remained in the cells for at least 48 h (Table 2, Appendix A2). However, after 5 days, most of the cell population returns back to QA3 (91%) (Appendix A2), similar to untreated cells (98% in QA3, Appendix A1), indicating that K-NDs exit the cells almost entirely in 5 days.

Like the cellular uptake, the addition of histidine onto the surface impacted the exit of NDs. In contrast to K-NDs that maintained residence in the cells for 48 h, the exit of histidine-containing NDs was much faster. The HK-NDs showed a slight decline in the population in QA1 from 21% to 17% after 48 h, indicating the exit of some internalized entities (Table 2; Appendix A3). H_50_K_50_-NDs showed a more pronounced decline in SSC, indicated by the highest downward shift from QA1 to QA3 after 48 h. After 48 h, the cell population in QA1 reduced from 49% to 23%, indicating a strong reduction in SSC intensity due to a reduced amount of light-diffracting H_50_K_50_-NDs in the cells. This indicates that the presence of histidine on the ND surface not only increases the cellular uptake but also possibly speeds up the cellular exit of NDs by at least twofold (Table 2; Appendix A4). Similarly to K-NDs, the histidine-based NDs show a marked recovery, leaving 10% or less of the cells in the QA1 quadrant 120 h post-treatment, showing that fNDs are lost from the cells after 5 days of exposure irrespective of functionalization.

### 3.5. Transfection Efficiency—GFP Knockdown

We assessed the contribution of histidine on the diamond surface to improve the transfection efficiency of fNDs. HeLa cells expressing GFP were treated with anti-GFP siRNA, and the degree of gene knockdown was recorded as a function of reduction in the fluorescence intensity of cells. Appendix A depicts a pre-treatment state at which the cells express high fluorescence (population tilted towards high GFP regions). A shift from the high-GFP region to low- and no-GFP region was monitored after delivering anti-GFP siRNA via different fNDs or lipofectamine RNAimax (a commercial standard for transfection used as the positive control). The cell population treated with the lipofectamine/anti-GFP-siRNA lipoplexes shifted almost entirely towards the low- and no-GFP region (Appendix A); therefore, the relative efficiency of diamoplexes to knockdown GFP fluorescence was calculated in relation to lipofectamine. Figure 8 shows the relative fraction of GFP knockdown induced by diamoplexes with respect to the lipofectamine control.

K-NDs showed the least GFP knockdown among all fNDs, i.e., 0.637 ± 0.02 (Figure 8 and Appendix A). The addition of histidine to the surface of NDs showed a significant increase in the fraction of GFP knockdown, i.e., 0.816 ± 0.05, at a similar mass ratio of 50:1 (Figure 8 and Appendix A).

We also evaluated the effect of reducing histidine on the surface to 50% in H_50_K_50_-NDs on the transfection efficiency at mass ratios of 30:1, 50:1, and 70:1. This was intended to determine the optimum balance needed to protect, carry, and release the siRNA in target cells. Diamoplexes of H_50_K_50_-NDs/anti-GFP siRNA at a 30:1 mass ratio showed the highest fraction of GFP knockdown among all diamoplexes, i.e., 0.830 ± 0.07 (Figure 8 and Appendix A). H_50_K_50_-NDs-based diamoplexes with anti-GFP siRNA prepared at a 50:1 mass ratio showed less GFP knockdown (0.593 ± 0.10) in comparison to the lower mass ratio of 30:1 (Figure 8 and Appendix A). Moreover, a higher mass ratio of 70:1 did not induce siRNA transfection at all, reducing the fraction of GFP knockdown to 0.295 ± 0.03, similar to the negative control (Figure 8 and Appendix A).

Lastly, we also confirmed that reduced GFP fluorescence results from transfection of anti-GFP siRNA as the scrambled siRNA delivered under the same conditions via K-NDs, HK-NDs and H_50_K_50_-NDs showed no GFP knockdown (fraction of GFP knockdown = 0.102, 0.297 and 0.159, respectively) (Figure 8).

## 4. Discussion

The intracellular fate of NPs critically controls the efficiency of a nucleic-acid delivery carrier. An efficient carrier is defined by its ability to protect the genetic material during circulation and against intracellular defense mechanisms. Events encountered by NP/nucleic acid complexes within the cells depend on the uptake mechanisms. Our previous studies show that fNDs internalize mammalian cells via two endocytic mechanisms: clathrin-mediated endocytosis and macropinocytosis, the latter being the most prominent route for diamoplexes [26]. We found that, regardless of the internalization pathway, both NDs alone and diamoplexes ended up in vesicular structures, anticipated as endosomes, which later became darker due to accumulation of degradative enzymes as it transitioned into lysosomes [26]. Enzymatic breakdown of the genetic cargo carried by these diamoplexes within the endo-lysosomal system was thus identified as the major limiting factor impeding the success of the K-NDs as gene carriers.

Therefore, this work focused on equipping the NDs with a machinery capable of triggering endo-lysosomal escape. There are many strategies suggested in the literature to release NP/nucleic acid complexes from the endosomes. We selected the addition of histidine in a relatively simple configuration on the surface for several reasons. The major criterion was to maintain the optimum size and positive surface charge. Unlike non-functionalized NDs, the K-NDs showed a unimodal size distribution, with most particles in the size range of 30–90 nm and zeta potential of +26.3 mV [23]. Therefore, the aim was to maintain the particle size below 200 nm [42]. A second important criterion was to limit aggregation and maintain dispersion stability. Lysine-functionalization of NDs minimized aggregation and maintained dispersion stability for one month. Uniform positive charges on the surface capable of maximizing electro-repulsive forces between particles, conferred by a zeta potential of more than 20 mV, were responsible for such an effect [23]. New functional moieties introduced on the surface should not disturb this equilibrium of charges to maintain optimum dispersion stability. It was also important to maintain optimal gene binding affinity. K-NDs have primary amines on the surface capable of binding negatively charged genetic material via electrostatic interactions, enabling the formation of diamoplexes at a 20:1 mass ratio. Therefore, the new strategy aimed to maintain this charge balance to allow optimal gene binding at a mass ratio conferring to non-toxic concentrations. Our assessments of the K-NDs indicated that they are biocompatible at a wide range of concentrations. In addition, they are stable in the biological media. The strategy adopted for optimizing the design should therefore target simple biomolecules, preferably amino acids, to maintain this pre-designed chemistry of the surface. Lastly, we attempted to maintain the hydrophilicity of the system. Lysine being a basic amino acid imparts good water dispersibility to the diamoplexes. This will ensure stability in the hydrophilic environment of the cell.

In order to meet the requirements mentioned above, we decided to modify the NDs using basic amino acids capable of inducing the osmotic instability of endosomes. Protonation occurs; hence, the endosomal membrane rupture is dependent on the acid-base balance of the amino acids. Lysine, histidine and arginine have ionizable amines in their hydrophilic side chain, determining the protonation state of amino acids at physiological pH. If the pH >> pKa, the amines will be neutral. If the pH << pKa, the amines will become protonated. Due to the pKa values of 8.95 and 10.53, both amine groups on the lysine will be protonated (side chain around 99%) at a physiological pH of 7.4. There is no evidence in the literature that protonated lysine triggers a proton sponge effect in the endosomes. Arginine behaves similarly due to even higher pKa values (9.04 and 10.53). In contrast to lysine and arginine, the pKa values of histidine are 9.17 and 6. The ionizable amine in imidazole side chain of histidine will thus show only 10% protonation at physiological pH maintained during formulation, blood circulation and cellular localization. However, upon entering the early sorting endosomes (pH 5.9–6.0), this amine will start to protonate, which further progresses during transition from early to late endosomes (pH 5.0–6.0). As mentioned above, protonation can eventually disturb the osmotic balance of the organelle, causing the influx of counter ions and water, causing endosomal swelling and membrane rupture before lysosomal fusion and degradation, as illustrated in Figure 9.

The addition of histidine was planned in two configurations: HK-NDs (having 100% surface coverage with histidine) and H_50_K_50_-NDs (having 50% surface coverage with lysine and 50% with lysyl-histidine). The number of histidine moieties on the ND surface was a critical parameter of control to maintain an optimum balance between four important characteristics of the carrier: (i) physicochemical stability of the dispersion, (ii) gene binding affinity, (iii) biological safety, and (iv) nucleic acid transfection efficiency. HK-NDs showed a comparable particle size distribution and zeta potential to the K-NDs, thus maintaining physicochemical characteristics of the system. However, for the H_50_K_50_-NDs, a two-fold dilution was needed (to 1 mg/mL) to maintain the dispersion stability of NDs. This could be attributed to changing the packing of lysine and lysyl-histidine moieties conjugated on the surface. Stearic hindrance posed by the bulkiness and longer chain length of lysyl-histidine conjugates may have impeded complete surface functionalization, leaving unfunctionalized carboxylic acids on the surface. In a concentrated dispersion, increased inter-particle interactions could lead to aggregation, disturbing the overall dispersion stability. Due to the fact that ND dispersions are not homogenous and may show loose aggregation owing to inter-particle interactions, we considered both the volume distribution and average particle size to compare dispersion characteristics. We found that the percent volume distribution of particles provided the best representation of the dispersion stability.

Assessment of the gene-binding efficiency of NDs after histidine modification prompted future modifications. Attaching lysyl-histidine onto the NDs resulted in the need for a higher mass ratio to bind siRNA, namely 60:1 for the HK-NDs (Figure 5) vs. 20:1 for the K-NDs [23]. Lysine electrostatically binds with genetic material [43,44,45] and the K-NDs show efficient nucleic acid binding [15,23]. Therefore, in the second histidine-containing derivative, we wanted to also expose free lysine on the surface. To accomplish that, NDs were functionalized with lysine and lysyl-histidine moieties in a 1:1 ratio, forming H_50_K_50_-NDs. The strategy to increase free lysine and reduce histidine to 50% on the ND surface improved the affinity of fNDs for siRNA and pDNA up to fivefold (Figure 4 and Figure 5). Compared to the K-NDs, H_50_K_50_-NDs improved the binding affinity to siRNA and pDNA by at least twofold.

We also compared the biocompatibility profiles of the NDs after the addition of histidine. Hard NPs such as zinc oxide, silica and gold with a positive surface showed higher cytotoxicity in non-phagocytic cells as compared to their negative counterparts of similar size [46]. In contrast, K-NDs do not induce cytotoxicity (Figure 6 and Figure 7). Unlike K-NDs, HK-NDs reduced the cell viability to <75% even at therapeutic concentrations, showing that the addition of histidine throughout the surface compromised the innately biocompatible nature of the diamonds. A reduction in biocompatibility for HK-NDs could be attributed to the presence of secondary amines on the surface. This is in correlation with literature showing that a high presence of secondary amines in the structure increases the cellular toxicity of polymers as well [47]. Linear analogues in the polymers with a higher amount of secondary amines were more cytotoxic than their branched counterparts. As the amount of secondary amines reduced in the branched structure due to the formation of amide bonds from the branching units, the cell viability improved significantly by 20% (fraction of cell survival was 60% and 80% for linear and branched polymers, respectively) [48]. Another study comparing the safety of quantum dots reported that quantum dots functionalized with secondary amines triggered the release of pro-inflammatory chemical mediators such as tumor necrosis factor-α [48]. We observed a similar trend for histidine-functionalized NDs. As the amount of histidine reduced to 50% on the surface in H_50_K_50_-NDs, the cell viability improved (Figure 6 and Figure 7), restoring the biocompatible character of fNDs.

We evaluated the cellular uptake and residence profiles of all fNDs. Overall, the addition of histidine on the surface enhanced the cellular uptake of NDs. However, unlike K-NDs, the residence of histidine-modified NDs was shorter than 48 h in the cells. To assess the cellular uptake and exit profiles, we capitalized on the fact that after ND uptake there was an increase in granularity and internal complexity in the treated cells owing to the presence of the light refracting features in the cytoplasm. These changes can be attributed to either the internalized NPs or increased activation of endosomes and lysosomes for trapping the internalized NPs [49]. As NDs possess a high refractive index (n = 2.42) due to the sp^3^ hybridized core [50,51], they are efficient light-scattering entities. Our group developed flow cytometric analysis of SSC patterns earlier to evaluate the dose-dependent cellular internalization of NDs [15,52]. The H_50_K_50_-NDs were internalized by the cells by the highest amount, followed by K-NDs (Appendix A and Table 2). Despite the fact that both HK-NDs and H_50_K_50_NDs exhibit a highly positive surface and average particle size in the colloidal range, we found that the presence of free lysine on the surface of H_50_K_50_NDs facilitated cellular uptake (Appendix A). This finding is in line with literature elucidating the use poly-lysine and other lysine-based constructs to improve the cellular internalization of various delivery platforms [53]. In addition to the cellular uptake, the exit of NDs from the cells was also studied. Elimination of NDs from cells is an important aspect of biosafety of NPs; however, the literature is scarce regarding the residence times of hard NPs in the cells. Similar to the uptake, the H_50_K_50_-NDs were also released quicker, starting after 48 h as a cellular population in the quadrant showing cells with internalized diamonds reduced to almost half at this time-point (Appendix A and Table 2).

Gene knockdown was achieved after 48 h of exposure to both HK-ND and H_50_K_50_-ND based diamoplexes (Figure 8); therefore, the elimination of these fNDs from the cells after 48 h of exposure is not expected to hinder the therapeutic outcome. Overall, all fNDs exited the cells after 5 days irrespective of functionalization (Appendix A and Table 2). These results were similar to amine-functionalized iron oxide NPs found in vesicular structures in the cytoplasm that were efficiently secreted from the cells 6 days after intracellular delivery [54]. Another work studying NDs showed the same trends as our evaluations. Flow cytometric assays combined with cell doubling-time measurements indicate that most NDs exit HeLa cells within 6 days of internalization, leaving ≤15% internalized diamonds at this time point [55]. Overall, the literature shows that the process of cellular exit is dependent upon the particle size of internalized entities. The smaller the particle size, the faster the exit process [54,56]. For example, the 15 nm-sized amine-functionalized iron oxide NPs highlighted above left the cells faster than their 30 nm counterparts, indicating that a small particle size favours cellular exit [54]. Apart from physicochemical NP factors, biological parameters such as the cell type and incubation time also impact the residence period [57,58].

Following studies of the fate of NDs, we compared the transfection efficiency of fNDs after histidine modification. Overall, our assessment revealed that histidine-modified ND constructs were able to transfect anti-GFP siRNA better than the prototype K-NDs (Figure 8). H_50_K_50_-NDs were superior to HK-NDs in terms of transfection efficiency as they show similar levels of gene knockdown at a lower mass ratio (30:1 vs. 50:1) (Figure 8). Interestingly, a higher mass ratio of 70:1 did not induce any gene knockdown (Figure 8), indicating that an optimum ratio of NDs to siRNA is critical to ensure efficient packaging, protection and release of the genetic cargo intracellularly. We used lipofectamine RNAimax as the gold standard to compare the transfection efficiencies of fNDs. Although a lipid-based positive control is not ideal, there is no commercial solid-core transfection agent to our knowledge. Additionally, although lipofectamine induced the highest siRNA-mediated GFP knockdown, the clinical application of this carrier is limited to due its cytotoxic nature [59]. As lipofectamine RNAimax is known to compromise the viability of cells [59], we compared gene transfection efficiencies by analysing an equal number of events in the healthy cell population. This way, we ensured that the fluorescence shift from a high to low GFP region is due to siRNA-mediated GFP knockdown rather than cell death. Although K-NDs induced siRNA transfection, our results provide evidence that the introduction of the pH-sensitive histidine moiety in the functionalization serves as a promising approach to improve intracellular trafficking of diamoplexes. This finding is in alignment with previous evidence in the literature. Modification of cationic lipid via covalent grafting of a single histidine improved the cytosolic transfer of DNA from endosomes, causing an increase in the overall transfection efficiency of the carrier (approximately eight times higher luciferase activity measured in RLU (%)) [60].

Our studies conclude that the design of H_50_K_50_-NDs exhibits a good balance between all four aspects of an efficient carrier as mentioned above. While the K-NDs showed good colloidal stability and nucleic acid binding, the intracellular release of anti-GFP siRNA was insufficient. Our current research is focused on utilizing functional siRNA that targets specific cellular processes in diseased cell models to demonstrate the translatability of the platform to human diseases. Future endeavours using new ND designs will target overexpressed genetic processes in cancerous cells. Biodistribution analysis performed on fNDs revealed that histidine-modified NDs largely accumulated in the liver and maintain a long residence time of at least 72 h [39]. In light of this finding, we aim to examine the transfection efficiency of histidine-modified constructs by targeting oncogenes in human hepatoma model.

## 5. Conclusions

Conclusively, this work provides a proof of concept that the addition of histidine in the design of amino acid/peptide fNDs is a promising approach to improve the overall performance of the nucleic acid carrier. Histidine acts as an endosomal membrane destabilizer and may facilitate the release of diamoplexes from early endosomes, thus overcoming the major rate limiting step in ND-mediated gene transfection. The addition of histidine in both configurations (100% and 50% surface coverage) maintains a particle size distribution in the colloidal range, surface charge and dispersion stability, similarly to K-NDs. However, the amount of histidine on the surface requires stringent control and optimization, as a higher amount is associated with reduced nucleic acid binding and increased cytotoxicity at cellular levels. The formation of H_50_K_50_-NDs where lysine and lysyl-histidine conjugates are at a 50/50 conformation is a good start to regain nucleic acid binding affinity and biocompatibility of the carrier and also to improve the transfection efficiency. Future work will be directed to applications of functional siRNA diamoplexes targeting overexpressed oncogenes in cancerous cells.

## Figures and Tables

**Figure 1 pharmaceutics-14-01794-f001:**
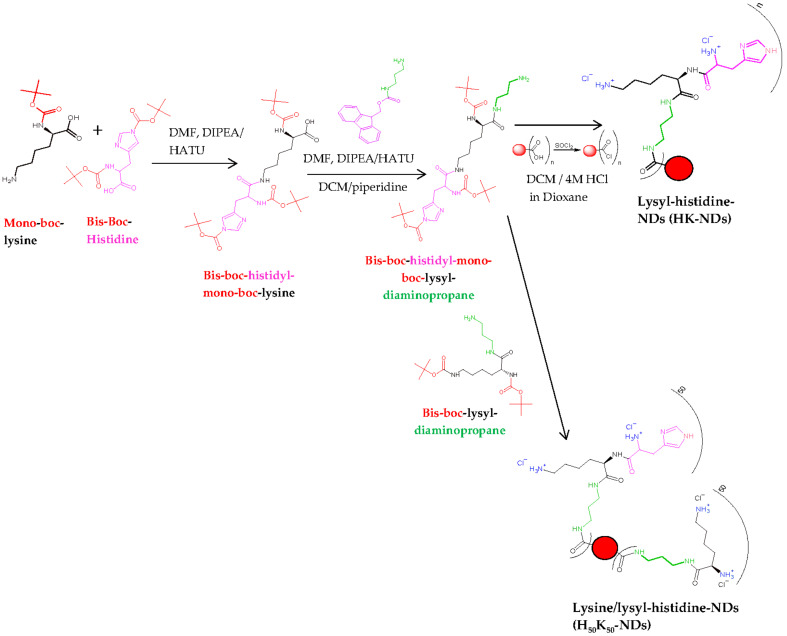
Schematic illustration to design lysyl-histidine-NDs (HK-NDs) and lysine/lysyl-histidine-NDs (H_50_K_50_-NDs). Final structures represent charged states of HK-NDs and H_50_K_50_-NDs at physiological pH.

**Figure 2 pharmaceutics-14-01794-f002:**
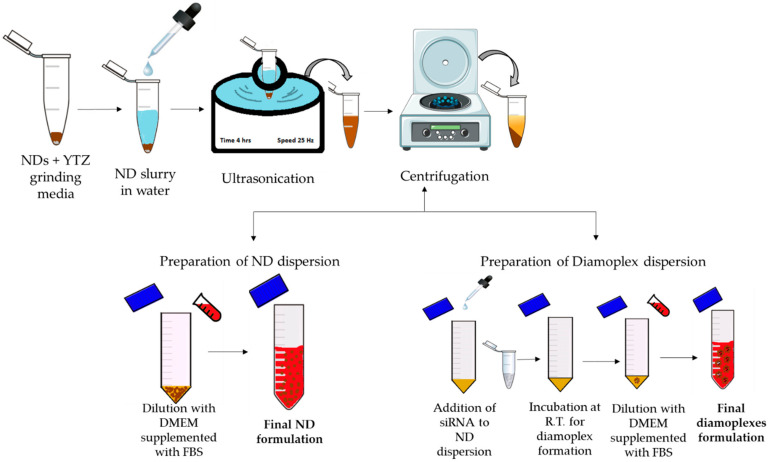
Schematic illustration of developing final formulation of NDs and diamoplexes for in vitro bioassays. RT = room temperature; DMEM = Dulbecco’s modified eagle’s medium; FBS = fetal bovine serum.

**Figure 3 pharmaceutics-14-01794-f003:**
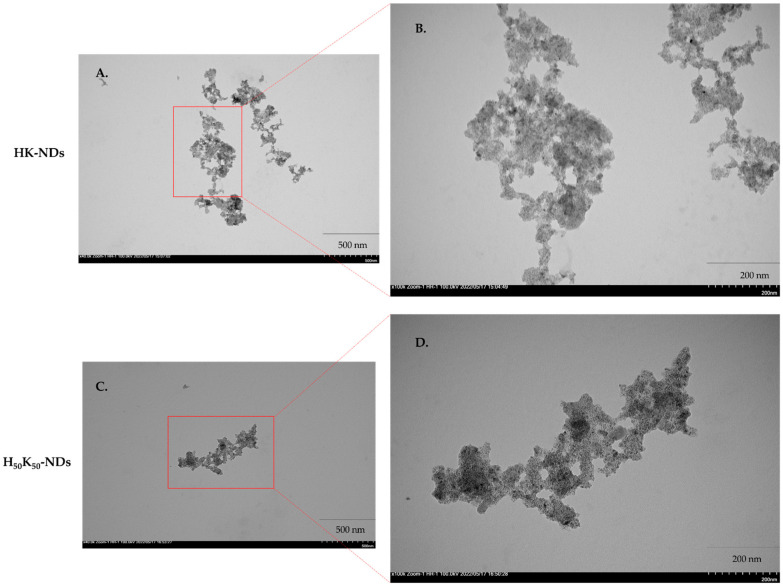
Electron micrographs of HK-NDs (top) and H_50_-K_50_-NDs (bottom) taken at 40,000× magnification (**A**,**B**) and 100,000× magnification (**C**,**D**). Bar on the lower right corner of the micrographs represent 500 nm (**A**,**B**) and 200 nm (**C**,**D**). Red boxes highlight portions of micrograph enlarged to show surface morphology of the clusters. Clear particle boundaries are visible for both HK-ND and H_50_K_50_-ND clusters. (HK-NDs = lysyl-histidine-NDs and H_50_K_50_-NDs = lysine/lysyl-histidine-NDs).

**Figure 4 pharmaceutics-14-01794-f004:**
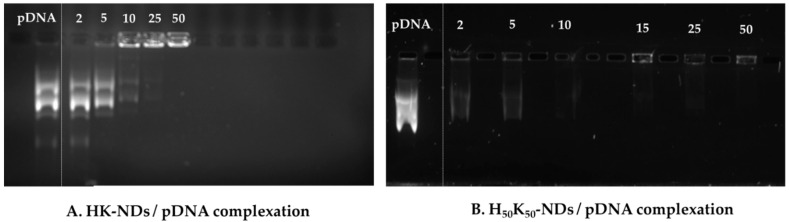
Binding of (**A**) HK-NDs and (**B**) H_50_K_50_-NDs with pDNA. HK-NDs binds with pDNA at minimum mass ratio of 10:1 while H_50_K_50_-NDs significantly improve gene binding starting to bind pDNA at a minimum mass ratio to 2:1. Although slight movement of band is visible at 2:1 and 5:1 mass ratios for H_50_K_50_-NDs, absence of strong bands as seen in control provides evidence of significant retardation of pDNA movement in the gel owing to strong electrostatic binding at these mass ratios (HK-NDs = lysyl-histidine-NDs; H_50_K_50_-NDs = lysine/lysyl-histidine-NDs; pDNA = plasmid deoxyribonucleic acid).

**Figure 5 pharmaceutics-14-01794-f005:**
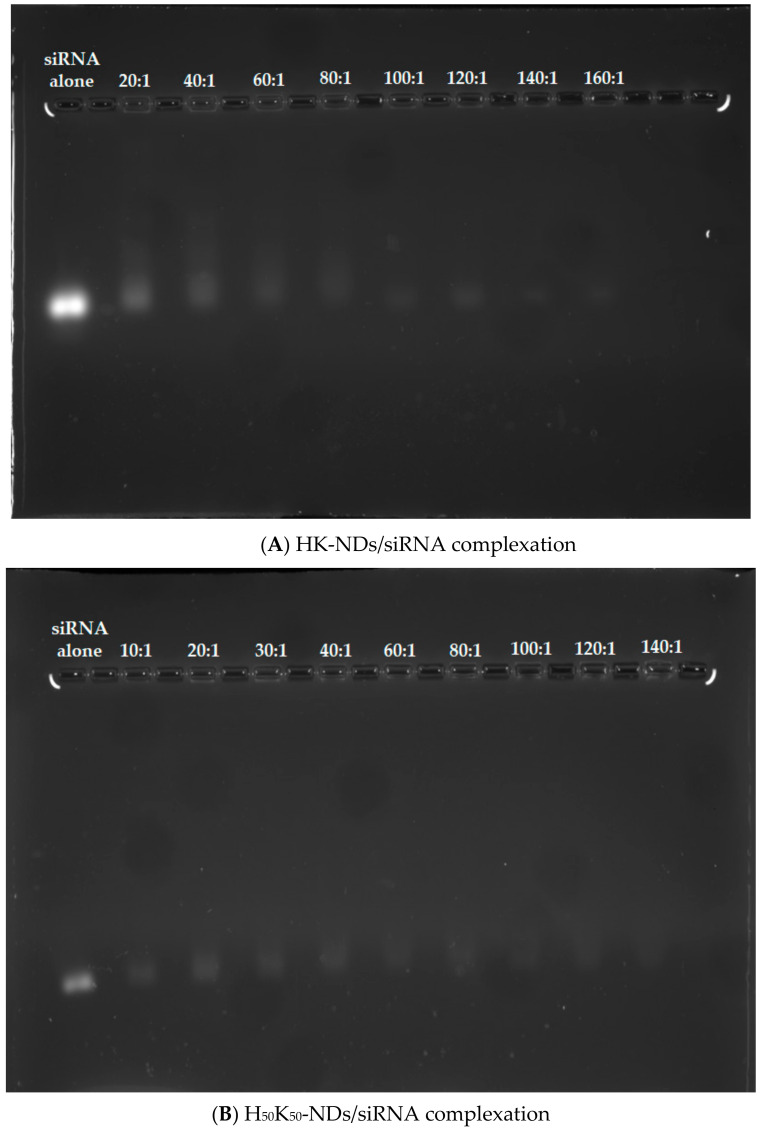
Binding of **(A**) HK-NDs and (**B**) H_50_K_50_-NDs with siRNA. HK-NDs binds with siRNA at minimum mass ratio of 60:1 while H_50_K_50_-NDs significantly improves gene binding, reducing the minimum mass ratio to 30:1 (HK-NDs = lysyl-histidine-NDs; H_50_K_50_-NDs = lysine/lysyl-histidine-NDs; siRNA = small interfering ribonucleic acid).

**Figure 6 pharmaceutics-14-01794-f006:**
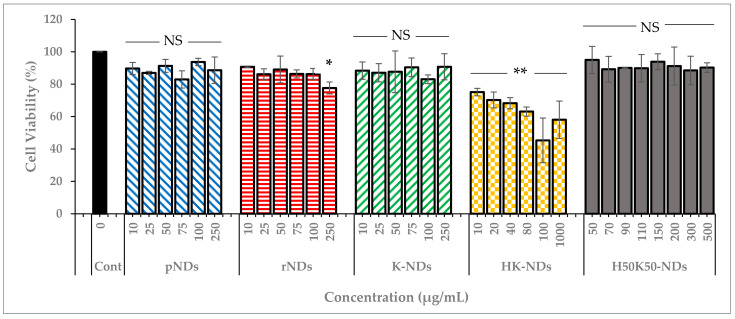
Cell viability after treatment with NDs. Carboxylating the surface of NDs forming rNDs did not cause a significant decrease in cell viability, remaining at 78%. K-NDs are also safe for the cells maintaining the cell viability at 91% at the highest concentration. A 100% surface coverage with histidine significantly affected the biocompatibility as the cell viability reduced to as low as 45% after HK-NDs at 100 µg/mL. Reducing the histidine to 50% led to recovery of the biocompatible nature of the NDs as H_50_K_50_-NDs did not reduce the viability by more than 10% at a concentration of 500 µg/mL. Statistical tests: two-way ANOVA and Tukey’s multiple comparison analysis were used. NS = not significant (*p*-value > 0.05); * significant with *p*-value = 0.010; ** significant with *p*-value < 0.001 (pNDs = pristine-NDs; rNDs = reoxidized NDs; K-NDs = lysine-NDs; HK-NDs = lysyl-histidine-NDs and H_50_K_50_-NDs = lysine/lysyl-histidine-NDs).

**Figure 7 pharmaceutics-14-01794-f007:**
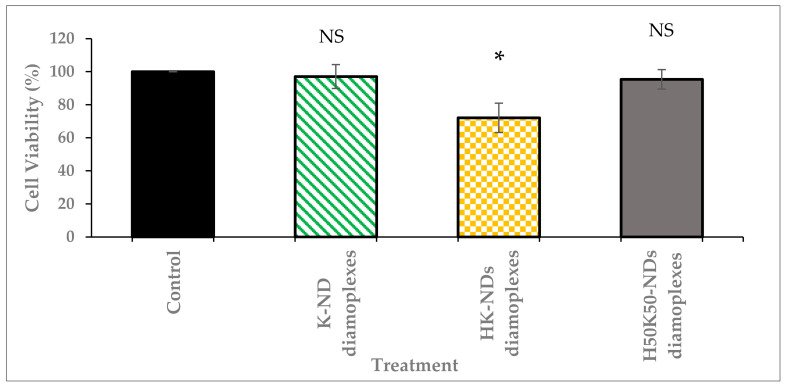
Cell viability after treatment with diamoplexes of K-NDs, HK-NDs and H_50_K_50_-NDs. Diamoplexes impact cell viability in the following order: K-ND ≈ H_50_K_50_-NDs < HK-NDs. * Diamoplexes were prepared as follows: K-NDs/siRNA diamoplexes at a mass ratio of 20:1, HK-NDs/siRNA diamoplexes at a mass ratio of 50:1, and H_50_K_50_-NDs/siRNA diamoplexes at a mass ratio of 30:1 (ND:siRNA). Mole ratios to form diamoplexes using each type of fND were dictated by their differential nucleic-acid binding ability. Statistical test: one-way ANOVA; Tukey’s multiple comparison analysis; NS = not significant (*p*-value > 0.05); * significant with *p*-value < 0.001) (K-NDs = lysine-NDs; HK-NDs = lysyl-histidine-NDs and H_50_K_50_-NDs = lysine/lysyl-histidine-NDs.

**Figure 8 pharmaceutics-14-01794-f008:**
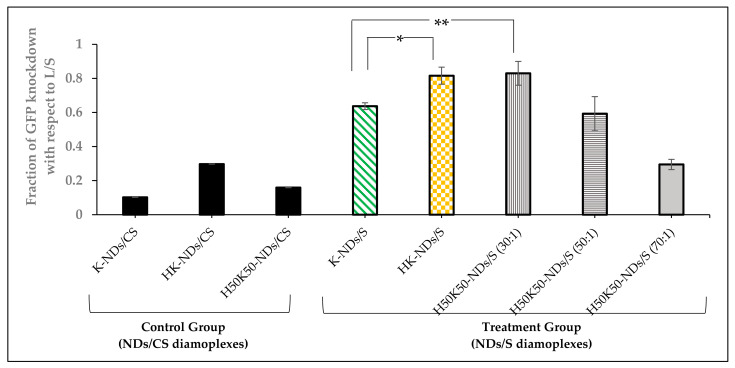
Fraction of GFP knockdown induced by diamoplexes of K-NDs, HK-NDs and H_50_K_50_-NDs with anti-GFP siRNA or scrambled siRNA with respect to lipofectamine RNAimax-based lipoplexes 48 h post-exposure. Fraction of GFP knockdown closer to 1 is representative of better siRNA transfection by diamoplexes. Statistical tests: one-way ANOVA; *p*-value < 0.001 and Tukey’s post hoc analysis showing that difference in mean is statistically significant between * K-NDs/S vs. HK-NDs/S (*p*-value = 0.019) and ** K-NDs/S vs. H_50_K_50_-NDs at 30:1 (*p*-value = 0.021) and not statistically significant between HK-NDs vs. H_50_K_50_-NDs at 30:1 and K-NDs vs. H_50_K_50_-NDs at 50:1 value. (NDs/CS = diamoplexes of functionalized NDs with scrambled siRNA used as the negative control; NDs/S- diamoplexes of functionalized NDs with anti-GFP siRNA; L/S = lipofectamine-based lipoplexes containing anti-GFP siRNA; K-NDs = lysine functionalized NDs; HK-NDs = lysyl-histidine functionalized NDs and H50K50-NDs = lysine/lysyl-histidine functionalized NDs).

**Figure 9 pharmaceutics-14-01794-f009:**
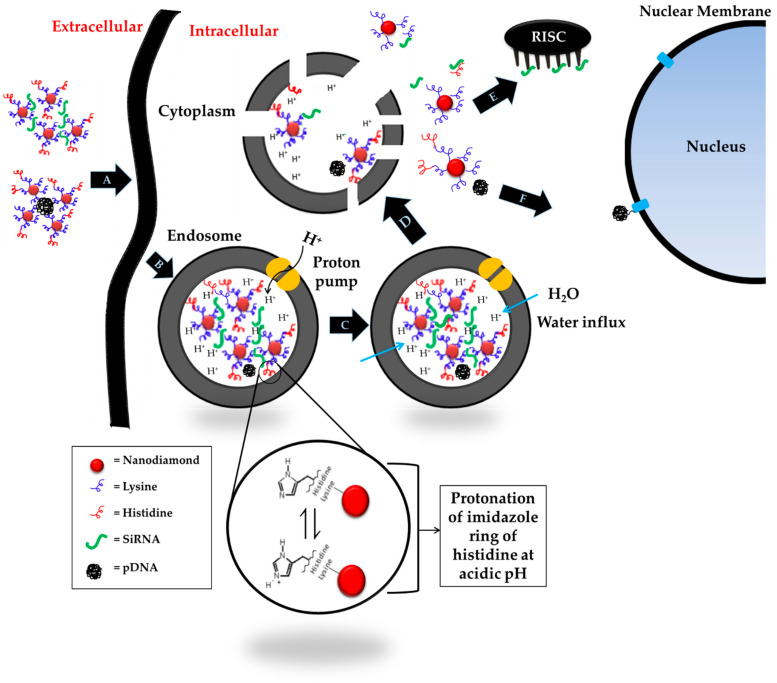
Schematic illustration of histidine-mediated endosomal escape. (**A**) Endocytosis of diamoplex by cell membrane, (**B**) formation of endosome, acidification and subsequent protonation of histidine, (**C**) disruption of osmotic balance within endosome by counter ion and water influx causing endosomal swelling, (**D**) rupture of endosomal membrane and release of diamoplex in the cytosol, (**E**) binding and expression of siRNA by RNA-induced silencing complex (RISC) in the cytoplasm, and (**F**) transfer of pDNA to the nucleus for expression by nuclear membrane transport proteins (p-DNA = plasmid deoxyribonucleic acid; siRNA = small interfering ribonucleic acid).

**Table 1 pharmaceutics-14-01794-t001:** Particle size distribution and zeta potential of HK-NDs and H_50_K_50_-NDs.

Sample	Volume Particle Size Distribution		Average Particle Size (nm)	Polydispersity Index (PDI)	Zeta Potential ± Standard Deviation (mV)
	Size Range (nm)	Percent Distribution (%)			
HK-NDs (2 mg/mL)	30–90	56	250	0.2	30 ± 3
	100–200	31			
	210–300	9			
H_50_K_50_-NDs (2 mg/mL)	<200	0	1949	0.1	18 ± 4
	210–500	1.7			
	510–1000	0.6			
	>1000	97.5			
H_50_K_50_-NDs (1.5 mg/mL)	<200	3.1	559	0.4	19 ± 6
	210–500	11			
	510–1000	38.1			
	>1000	48.8			
H_50_K_50_-NDs (1 mg/mL)	<200	50.5	194	0.2	23 ± 6
	210–500	30.8			
	510–1000	10.7			
	>1000	8.2			

Particle size distribution is measured in nm (nanometers) and is represented as percent volume. Size data is acquired as a mean of three measurements each comprising 10 individual runs. Zeta potential is measured in milli volts (mV) and represent a mean of three measurements each comprised of 100 individual runs. All dispersions of NDs were made in sterile molecular grade water. (HK-NDs = lysyl-histidine-NDs and H_50_K_50_-NDs = lysine/lysyl-histidine-NDs).

**Table 2 pharmaceutics-14-01794-t002:** Cellular internalization of NDs in after 24, 48 and 120 h post-treatment. Shift to QA1 corresponds to higher side scattering attributed to the internalized NDs.

Time-Point	Sample	Cells in QA3 (%)	Cells in QA1 (%)
24 h post treatment	Untreated	94.86	5.14
	K-NDs	75.75	24.25
	HK-NDs	79.27	20.73
	H_50_K_50_-NDs	51.45	48.55
48 h post-treatment	Untreated	95.78	4.22
	K-NDs	73.86	26.14
	HK-NDs	82.44	17.56
	H_50_K_50_-NDs	76.61	23.29
120 h post treatment	Untreated	98.79	1.21
	K-NDs	90.69	9.31
	HK-NDs	91.64	8.36
	H_50_K_50_-NDs	89.03	10.97

(K-NDs = lysine-NDs; HK-NDs = lysyl-histidine-NDs and H_50_K_50_-NDs = lysine/lysyl-histidine-NDs).

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
