# Peer review of "Design of Smart Nanodiamonds: Introducing pH Sensitivity to Improve Nucleic Acid Carrier Efficiency of Diamoplexes"

_pharmaceutics, 2022, doi:10.3390/pharmaceutics14091794_

Round 1
Reviewer 1 Report
The manuscript written by Alwani et al. is about designing and preparing nanodiamonds with the goal to be complexed with nucleic acids (plasmids and siRNA). The goal of this manuscript was to obtain functionalized nanodiamonds containing groups that facilitate endosomal escape. The resultant nanovehicles were fully characterized and their ability to silence gene expression was also studied together with cytotoxicity. On the whole, this manuscript is interesting, well organized and the data is convincing. I think this manuscript shows valuable information in the field of non-viral vehicles for the transport of siRNA oligonucleotides. Although my first impression of this manuscript is positive, the authors should consider the following suggestions to improve the quality of this manuscript:
Minor comments
(1) Page 4, line 136. Although the protocol is described elsewhere, the authors should improve the quality of the experimental part by adding the amount/concentration/mmol of reagents used for the synthesis.
(2) Page 8, line 246. The authors should include, if possible, the siRNA sequences (both passenger and guide) used in their study
(3) Figure 7. The authors stated that “diamoplexes were prepared with siRNA at ratios of 20:1, 50:1, and 30:1 for K-NDs, HK-NDs and H50K50-NDs, respectively.” I do not understand how the authors have displayed these data. If the authors have tested three different ratios for each complex, there should be more than three outcomes as displayed in the figure. Please, explain this controversy.
(4) Table 3. These results should be visualized better in the form of a bar chart rather than a table.
Major comments
(1) Figure 4. The authors show a gel electrophoresis assay to confirm at which ratio plasmid is able to complex with nanodiamonds and form the corresponding diamoplexes. The authors should improve this characterization by studying the DNAse I protection and SDS-induced release of DNA from diamoplexes.
(2) Figure 5. The authors have carried out a gel electrophoresis technique to visualize the complexation process of a siRNA with the nanodiamond. The authors should include another type of study that include: (i) a siRNA displacement assay and (ii) a serum stability assay.
(3) The manuscript should include confocal microscopy studies to study and confirm the cellular uptake of siRNA complexes
Reviewer 2 Report
A proof of concept was described by the authors where the addition of histidine on the surface of nanodiamond improves its overall performance as a nucleic acid carrier. The tweaking of the amount of histidine on the surface can compromise the nucleic acid binding and cytotoxicity at cellular levels. Nevertheless, several points need to be addressed before this manuscript can be considered for publication.
· In all the document ml should be replaced by mL.
· A schematic representation of the reactions involved should be illustrated in Figure 1, even though it follows the same protocol as ref 23.
· Since the amount of histamine moieties on ND surface is a critical parameter, the exact quantity should be determined.
· Is there a rational of picking 50% of histamine moieties on ND surface?
· Small typo error (line 146) 1.0 mmole/gm, should be 1.0 mmol/g
· Small incorrection line 176, agarose should be prepared in running buffer and no purified water
· What was used as a negative control? Should be referred. Line 196
· Figure 2, I do not understand what the blue boxes are.
· Cytotoxic assays should also contain a group of cells treated just with siRNA, without NPs.
· Clarification regarding why the dilution of the suspension influences the PDI, size and zeta. Ultimately should have no influence at all.
· TEM images of rNDs, and the scale bare is difficult to evaluate as well as the inserts.
· Don’t understand why the intensity of pDNA in figure 8 a and b are different since the same quantity was used.
· The statistics used in the cell viability assay are confusing, what is statistically different to what? Why the different tests were used.
· Figure 7 caption “K-ND << H50K50-NDs < HK-NDs”, should be K-ND ~H50K50-NDs < HK-NDs.
· In order to evaluate the uptake a fluorescent tag should be used. I am not convinced with the setting used. The QA3 gating should always be 95% or a fixed %. Figure 8 should go to supplement information. The same gate should be used regardless the timepoint.
· Transfection efficiency data treatment is not convincing. Histograms of treatment as well as Negative and positive controls should be given in supplement information.
· Just because the K-ND pathway is clathrin mediated endocytosis and micropinocytosis, does not mean HK-ND are. Evidence show be shown.
· Figure 9 the color scheme of the Nanodiamond is not the same in all the scheme.
Round 2
Reviewer 1 Report
I recommend this manuscript for publication in Pharmaceutics